# A powerful drug combination strategy targeting glutamine addiction for the treatment of human liver cancer

Haojie Jin[1,2†], Siying Wang[1†], Esther A Zaal[3†], Cun Wang[1,2], Haiqiu Wu[4], Astrid Bosma[2], Fleur Jochems[2], Nikita Isima[2], Guangzhi Jin[5], Cor Lieftink[2], Roderick Beijersbergen[2], Celia R Berkers[3,6*], Wenxin Qin[1*], Rene Bernards[2*]

[1]State Key Laboratory of Oncogenes and Related Genes, Shanghai Cancer Institute, Renji Hospital, Shanghai Jiao Tong University School of Medicine, Shanghai, China; [2]Division of Molecular Carcinogenesis, Oncode Institute. The Netherlands Cancer Institute, Amsterdam, Netherlands; [3]Biomolecular Mass Spectrometry and Proteomics, Bijvoet Center for Biomolecular Research, Utrecht University, Utrecht, Netherlands; [4]Department of Cell and Chemical Biology, Leiden University Medical Centre, Leiden, Netherlands; [5]Department of Pathology, Eastern Hepatobiliary Surgery Hospital, Second Military Medical University, Shanghai, China; [6]Department of Biochemistry and Cell Biology, Faculty of Veterinary Medicine, Utrecht University, Utrecht, Netherlands

*For correspondence:
C.R.Berkers@uu.nl (CRB);
wxqin@sjtu.edu.cn (WQ);
r.bernards@nki.nl (RB)

†These authors contributed equally to this work

Competing interests: The authors declare that no competing interests exist.

**Abstract** The dependency of cancer cells on glutamine may be exploited therapeutically as a new strategy for treating cancers that lack druggable driver genes. Here we found that human liver cancer was dependent on extracellular glutamine. However, targeting glutamine addiction using the glutaminase inhibitor CB-839 as monotherapy had a very limited anticancer effect, even against the most glutamine addicted human liver cancer cells. Using a chemical library, we identified V-9302, a novel inhibitor of glutamine transporter ASCT2, as sensitizing glutamine dependent (GD) cells to CB-839 treatment. Mechanically, a combination of CB-839 and V-9302 depleted glutathione and induced reactive oxygen species (ROS), resulting in apoptosis of GD cells. Moreover, this combination also showed tumor inhibition in HCC xenograft mouse models in vivo. Our findings indicate that dual inhibition of glutamine metabolism by targeting both glutaminase and glutamine transporter ASCT2 represents a potential novel treatment strategy for glutamine addicted liver cancers.

## Introduction

Liver cancer is one of the most common malignant tumors worldwide and ranks as the fourth leading cause of cancer death (*Bray et al., 2018*). Due to the rapid progress and delayed diagnosis, most liver cancer patients are diagnosed at an advanced stage, which leaves very few treatment options and poor prognosis. Currently approved agents, such as sorafenib and lenvatinib, provide only modest survival benefits to hepatocellular carcinoma (HCC) patients with low response rates (*Llovet et al., 2008*; *Kudo et al., 2018*). Therefore, there is an urgent need for novel therapies that target new critical dependencies of liver cancer.

Metabolic reprogramming contributes to tumor development and introduces metabolic liabilities that can be exploited to treat cancer. Despite significant efforts to develop drugs targeting metabolic pathways of cancer, clinical success has been very limited for decades. One particular potential vulnerability of cancer cells is the metabolic reprogramming toward aerobic glycolysis, known as the

Warburg effect (*Ward and Thompson, 2012*). However, the clinical use of one compound known to block glycolysis, 2-deoxyglucose (2-DG), is limited by toxicity and insufficient inhibition of tumor growth at tolerable doses (*Raez et al., 2013*). Recent studies have highlighted a very prominent contribution of glutaminolysis to energy and macromolecule homeostasis in several types of hematological and solid tumors (*Spinelli et al., 2017*; *Vander Heiden and DeBerardinis, 2017*). Glutaminolysis converts glutamine to glutamate, which is catalyzed by mitochondrial glutaminase (GLS), and then further converted into tricarboxylic acid (TCA) cycle metabolites, generating ATP and NADPH. Glutaminolysis is also directly involved in the regulation of reactive oxygen species (ROS) homeostasis by providing not only precursors glutamate and cysteine for glutathione (GSH) synthesis but also promoting the production of NADPH via glutamate dehydrogenase (GLUD; *Altman et al., 2016*). Many tumor types, such as pancreatic cancer (*Son et al., 2013*), acute myeloid leukemia (AML; *Jacque et al., 2015*), breast cancer (*Gross et al., 2014*), and lung cancer (*Wang et al., 2018*), are particularly dependent on glutamine for proliferation and survival, as glutamine controls oxidative phosphorylation, nucleotide biosynthesis, and redox homeostasis in these cells, and removal of glutamine leads to apoptosis. Recently, it is reported that GLS1, one major isoform of GLS, regulates the stemness properties of HCC, and targeting GLS1 achieved some therapeutic effect against HCC cells (*Li et al., 2019*).

CB-839 (also known as telaglenastat) is a potent and noncompetitive allosteric GLS1 inhibitor. It exhibits significant anti-proliferative activity in several types of cancer cell lines and xenografts such as triple-negative breast cancer (*Gross et al., 2014*), lung adenocarcinoma (*Galan-Cobo et al., 2019*), chondrosarcoma (*Peterse et al., 2018*), and lymphoma cancer (*Xiang et al., 2015*). CB-839 is used increasingly in combination to treat cancer. CB-839 overcomes metabolic adaptation to the mTOR inhibitor MLN128 in xenografts and patient-derived xenografts (PDXs) of lung squamous cell carcinomas (*Momcilovic et al., 2018*). Moreover, the combination of CB-839 and CDK4/6 inhibitors has shown promising results in human esophageal squamous cell carcinoma (*Qie et al., 2019*). CB-839 is also under evaluation for the treatment of hematological malignancies and solid tumors in several phase I-II clinical trials (*Song et al., 2018*). However, it is still unclear whether CB-839 has therapeutic potential in liver cancer. Here we show that CB-839 monotherapy is insufficient for the treatment of HCC and identify a novel combination strategy that effectively targets glutamine addiction in liver cancer.

## Results

### Liver cancer is addicted to glutamine

To determine whether liver cancer is a glutamine (Gln)-dependent tumor type, we cultured a panel of liver cancer cell lines in a medium with 4 mM Gln or without Gln. Both long-term colony formation assay (*Figure 1a*) and short-term IncuCyte assay (*Figure 1b*) showed that Gln deprivation impaired the proliferation of most liver cancer cell lines (9 out of 11) in vitro. Accordingly, we defined these 9 cell lines, which need exogenous Gln for efficient proliferation, as Gln dependent (GD) cells, and the other two cell lines as Gln independent (GID) cells. To assess Gln dependency of liver cancer in patients, we analyzed Gln metabolism-related genes in The Cancer Genome Atlas (TCGA) named 'PENG_GLUTAMINE_DEPRIVATION_DN', whose expression is considered to be positively correlated with Gln metabolism. The heatmap shows the obvious upregulation of these genes in liver cancer tissues as compared to their corresponding non-cancerous liver tissues in TCGA cohort containing 50 paired HCC samples (*Figure 1c*). Gene set enrichment approach (GSEA) also showed positive enrichment of genes associated with Gln metabolism in both TCGA and Gene Expression Omnibus (GEO) data (*Figure 1d and e*). Moreover, SurvExpress survival analysis of 381 TCGA samples indicated that the dysregulation of Gln metabolism genes correlated with poor prognosis of human liver cancer patients (*Figure 1f*). Taken together, these results indicate that the liver cancers of poor prognosis have upregulated glutamine metabolism, which may indicate an increased requirement for Gln in HCC tumors.

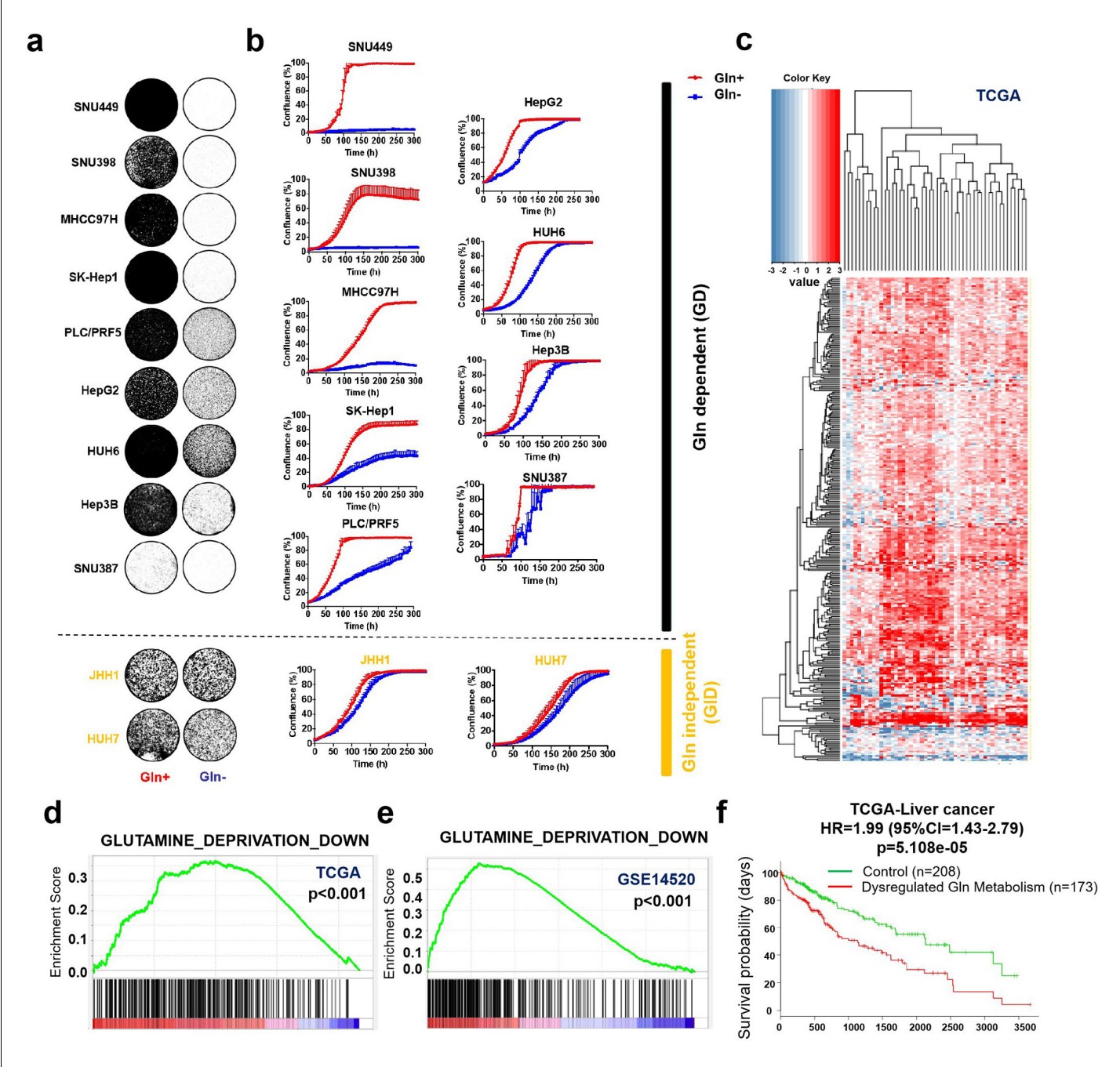

**Figure 1.** Liver cancer is addicted to glutamine. (a, b) A total of 11 liver cancer cell lines were cultured with 4 mM glutamine (Gln+) or glutamine deprivation (Gln-). Proliferation was assessed by colony formation assay (a) and IncuCyte assay (b), respectively. Liver cancer cell lines were divided into Gln dependent (GD) and Gln independent (GID) subtypes, respectively. (c) Differential expression profiles of Gln metabolism-related genes (Gene Set: PENG_GLUTAMINE_DEPRIVATION_DN) in 50 paired HCC samples in TCGA cohort. Heatmap illustrated the log2 fold-change values of Gln metabolism-related genes between cancerous tissues and their corresponding noncancerous liver tissues. Red color indicates gene upregulation; blue color indicates gene downregulation. Each column indicates a patient; each row indicates a gene. (d, e) Gene set enrichment analysis (GSEA) enrichment of cancerous tissues versus corresponding noncancerous tissues for gene set 'PENG_GLUTAMINE_DEPRIVATION_DN' in TCGA (d) and GSE14520, respectively (e). (f) Expression of Gln metabolism-related genes correlated with poor prognosis of liver cancer patients in TCGA. HR: hazard ratio; CI: confidence interval.

The online version of this article includes the following source data for figure 1:

**Source data 1.** Liver cancer is addicted to glutamine.

## The glutaminase inhibitor CB-839 monotherapy achieves insufficient anti-tumor effect in liver cancer

The glutaminase isoenzyme GLS1 is a key enzyme in Gln metabolism. We first analyzed the expression levels of GLS1 in the GSE14520 cohort (n = 229), which provides data on gene expression for both paired human non-tumor and HCC tissues. The results indicate that mRNA levels of GLS1 are significantly increased in HCC tissues compared to non-tumor tissues (*Figure 2a*). Among the 229 paired samples, the GLS1 level is increased to more than double in about 55% of HCC patients (*Figure 2b*). Next, we analyzed the association between GLS1 level and prognosis of liver cancer patients. TCGA data indicated that the high mRNA level of GLS1 correlated with the poor prognosis of human liver cancer patients (*Figure 2c*). We also analyzed GLS1 expression using a tissue microarray (TMA) containing 377 HCC specimens by IHC analysis. HCC patients were classified into two groups: GLS1$^{low}$ group (n = 175) and GLS1$^{high}$ group (n = 202). The Kaplan-Meier analysis indicates that HCC patients with high protein expression of GLS1 exhibit worse overall survival (OS) and disease-free survival (DFS) as compared to patients with low protein expression of GLS1 (*Figure 2d,e and f*). To explore the therapeutic effect of GLS1 inhibitor CB-839 on liver cancer cells, we treated the panel of 11 liver cancer cell lines with increasing concentrations of CB-839 in long-term colony formation assays and short-term CellTiter-Blue cell viability assays. We found that CB-839 treatment only severely impaired the proliferation of three GD cell lines (*Figure 2g and h*). Other six GD cell lines and two GID cell lines only showed little response to CB-839 treatment in vitro (*Figure 2g and h*). We also analyzed the correlation between CB-839 sensitivity and expression level of its target and found no correlation between protein level of GLS1 and CB-839 sensitivity (*Figure 2g, h and i*). In addition, the protein level of GLS1 was not correlated to Gln addiction in vitro (*Figures 1a, b* and *2i*), indicating that Gln addiction to liver cancer cells is not dependent on GLS1 level. These data suggest that targeting glutamine metabolism by GLS1 inhibitor CB-839 alone is insufficient for liver cancer therapy.

## A compounds screen identifies that ASCT-2 inhibitor V-9302 sensitizes GD liver cancer cells to CB-839 treatment

The data shown above indicate that a significant number of liver cancer cell lines are glutamine dependent but fail to respond to CB-839 treatment. To study this in more detail, we investigated metabolite profiles of two GD liver cancer cell lines, SNU398 and HepG2. A total of 66 named metabolites were identified and mapped to seven major pathways. We found that CB-839 treatment significantly decreased a number of key downstream metabolites involved in Gln metabolism, such as glutamate (GLU), TCA cycle intermediate (α-KG), redox metabolite (glutathione, NADPH) in both cell lines (*Figure 3a and b* and *Figure 3—figure supplement 1*). These results indicate that CB-839 efficiently blocks Gln utilization and interferes with the dynamic changes of intermediates in Gln metabolism. Therefore, we hypothesized that CB-839 treatment already caused metabolic vulnerability, which could further be exploited for cancer therapy if co-treated with other anti-metabolic drugs. To prove this, we generated a chemical library consisting of 13 compounds inhibiting a variety of tumor metabolism targets, and tested their ability to enhance the anti-tumor effect of CB-839. Notably, we found that V-9302, a novel inhibitor of Gln transporter ASCT2 (*Schulte et al., 2018*), is the most potent agent in sensitizing both SNU398 and HepG2 GD liver cancer cells to CB-839 (*Figure 3c and d*). To study whether this combination has a broad anti-proliferative effect in liver cancer cells, we tested cell viability and proliferation in a panel of liver cancer cell lines after single drug or combination treatment with CB-839 and V-9302 in vitro. Indeed, the combination showed synergistic anti-proliferation effect in GD cell lines, but only showed limited anti-tumor effect in GID cell lines in vitro (*Figure 4a,b and c* and *Figure 4—figure supplement 1*). Moreover, similar results were observed in these cell lines when combining V-9302 with another GLS1 inhibitor BPTES (*Figure 4—figure supplement 2*). These findings suggest that the combination of GLS1 inhibitors and V-9302 could be a novel therapeutic approach for GD liver cancer cells.

## Combination of CB-839 and V-9302 depletes glutathione and induces lethal ROS level in GD liver cancer cells

The alanine-serine-cysteine transporter, type-2 (ASCT2, encoded by gene *SLC1A5*), is a sodium-dependent solute carrier protein responsible for the import of neutral amino acids and is the primary

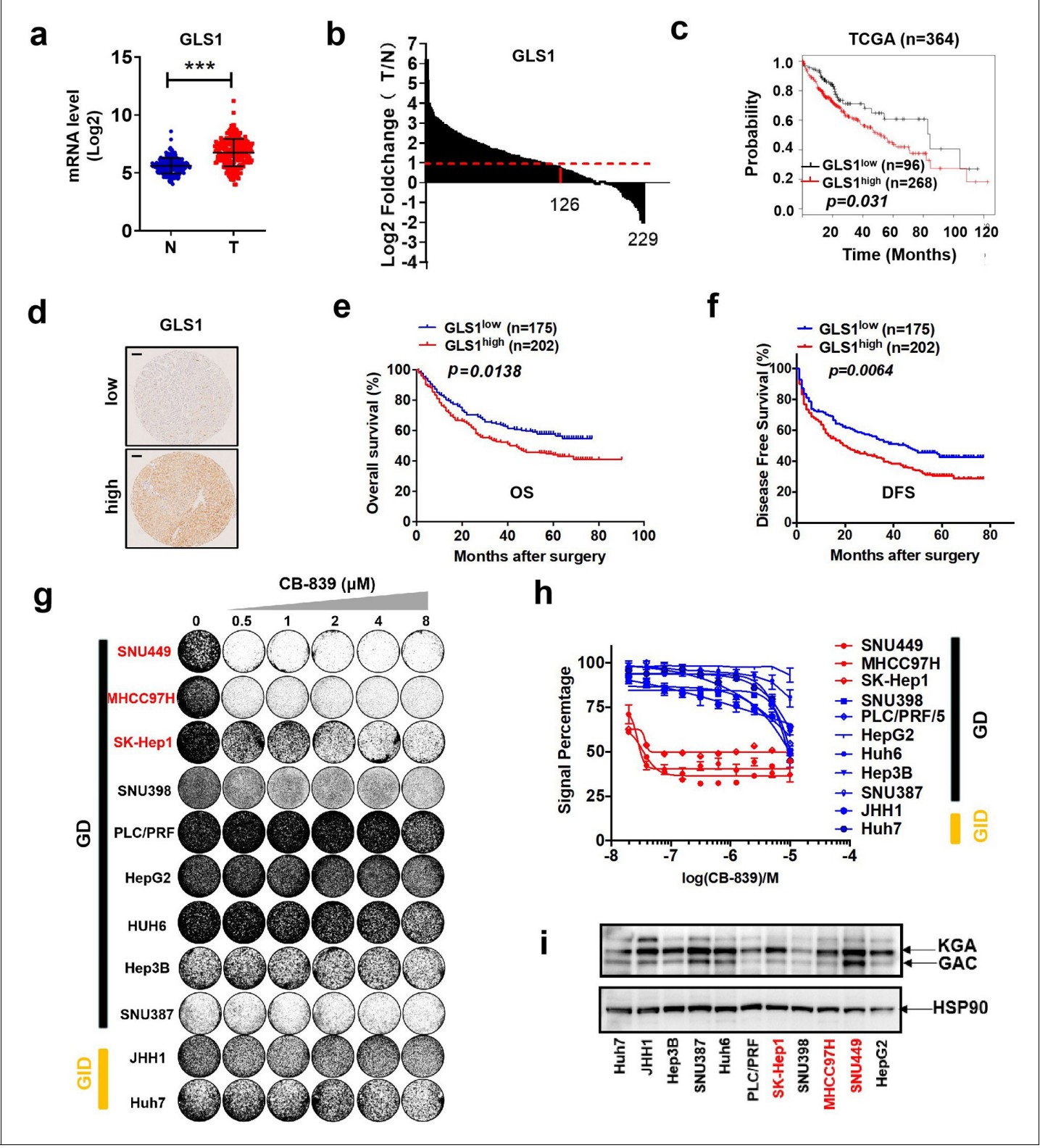

**Figure 2.** The glutaminase inhibitor CB-839 monotherapy shows an insufficient anti-tumor effect in liver cancer. (**a**) mRNA levels of GLS1 in the cohort of GSE14520 (n = 229; probe for GLS1: 203159_at; N: nontumor tissues, T: tumor tissues). Data are represented as mean ± SEM. (**b**) Log2 fold change of GLS1 mRNA in 229 paired HCC samples in the cohort of GSE14520 (probe for GLS1: 203159_at; N: nontumor tissues, T: tumor tissues). (**c**) GLS1 expression and Kaplan-Meier OS analysis for patients with HCC in TCGA cohort (n = 364). (**d–f**) IHC staining analyses of GLS1 were performed in 377 patients with HCC. The patients were divided into two groups: GLS1low group (n = 175) and GLS1high group (n = 202). (**d**) Typical immunostaining

*Figure 2 continued on next page*

*Figure 2 continued*

images of GLS1 in GLS1low group and GLS1high group were shown. Scale bars = 100 μm. The Kaplan-Meier analysis for OS (**e**) and DFS (**f**) was performed according to GLS1 levels. (**g, h**) Liver cancer cell lines were treated with increasing concentrations of CB-839. Proliferation and viability were assessed by colony formation assay (**g**) and CellTiter Blue assay (**h**), respectively. (**i**) Lysates of liver cancer cell lines were western blotted for two splice variants of GLS1 (KGA and GAC). HSP90 served as a control. ***p<0.001, Student's *t* test.

The online version of this article includes the following source data for figure 2:

**Source data 1.** The glutaminase inhibitor CB-839 monotherapy shows insufficient anti-tumor effect in liver cancer.

transporter of glutamine in cancer cells. Several studies have attributed glutathione (GSH) synthesis and ROS stress with dysregulation of ASCT2 (*Schulte et al., 2018*; *Yoo et al., 2020*). To investigate whether the combination of CB-839 and V-9302 can disrupt the ROS balance in liver cancer, we first analyzed the GSH levels after single-drug treatment or combination in SNU398 and HepG2 cells, respectively. The results show that CB-839 or V-9302 alone significantly decreased the level of GSH, while the combination resulted in a further decrease in GSH in both cell lines (*Figure 5a*). GSH is an important antioxidant that acts as a free radical scavenger upon its reaction with ROS in cells (*Bansal and Simon, 2018*; *Okazaki et al., 2017*), raising the possibility that combination of CB-839 and V-9302 may interfere with ROS homeostasis of these liver cancer cells. Analysis of intracellular ROS levels showed that single agent CB-839 or V-9302 only modestly increased the ROS production. However, their combination dramatically increased the already-elevated ROS production, reaching a level that caused severe DNA damage, as evidenced by an increase in γ-H2AX (*Figure 5b and c*). To determine whether the combination inhibited cell viability and proliferation of liver cancer cells via the excessive ROS production, we treated these cells with the ROS scavenger N-acetyl-l-cysteine (NAC). The results show that NAC treatment rescued the cell viability and proliferation of SNU398 and HepG2 cells in the presence of both CB-839 and V-9302 (*Figure 5d and e*). Moreover, the strong synergistic induction of apoptosis can be rescued by NAC treatment in both SNU398 and HepG2 cells, as indicated by the IncuCyte caspase-3/7 apoptosis assay (*Figure 5f*). These results suggest a model explaining the synergistic effect between CB-839 and V-9302 (*Figure 5g*): GLS1 inhibition by CB-839 reduces mitochondrial Glu level, which decreases TCA-dependent NADPH production, elevating glutathione oxidation. Besides, CB-839 also decreases intracellular Glu that is essential for cystine import by xCT, a cystine/glutamate antiporter, thus cutting down the cysteine conversion that in turn serves as the rate-limiting precursor for GSH biosynthesis. CB-839 treated cells are vulnerable to other perturbations that further deplete glutamine anaplerosis, such as blockage of major Gln transporter ASCT-2 by V-9302. Collectively, these results indicate that the combination of CB-839 and V-9302 achieves a synergistic anti-tumor effect in liver cancer via disrupting the GSH/ROS balance.

## Combined treatment inhibits xenograft growth and induces apoptosis in vivo

To assess the effectiveness of the combination of CB-839 and V-9302 in vivo, SNU398 and MHCC97H cells were injected into nude mice to establish tumors. After tumors reached a size of about 100 mm³, animals were treated with vehicle, CB-839, V-9302, or the combination of both drugs. Results showed that the combination elicited a strong growth inhibition in both SNU398 and MHCC97H xenograft models, while single-drug treatment showed modest anti-tumor effects (*Figure 6a,b,c and d*). We also measured the body weight of mice during the treatment, and no body weight reduction was observed in all the treatment groups (*Figure 6—figure supplement 1*), indicating good tolerability for this novel drug combination. IHC analyses showed that treatment with the combination of CB-839 and V-9302 resulted in an obvious decrease in Ki67 positive cells (*Figure 6e,f,g and j*). In addition, the combination also significantly increased caspase-3 positive cells (*Figure 6e,f,h and k*) and γH2AX-positive cells (*Figure 6e,f,i and l*) in tumor tissues, supporting the induction of apoptosis and DNA damage in vivo. Taken together, our xenograft model experiments point out that combination of CB-839 and V-9302 is effective for liver cancer therapy in vivo.

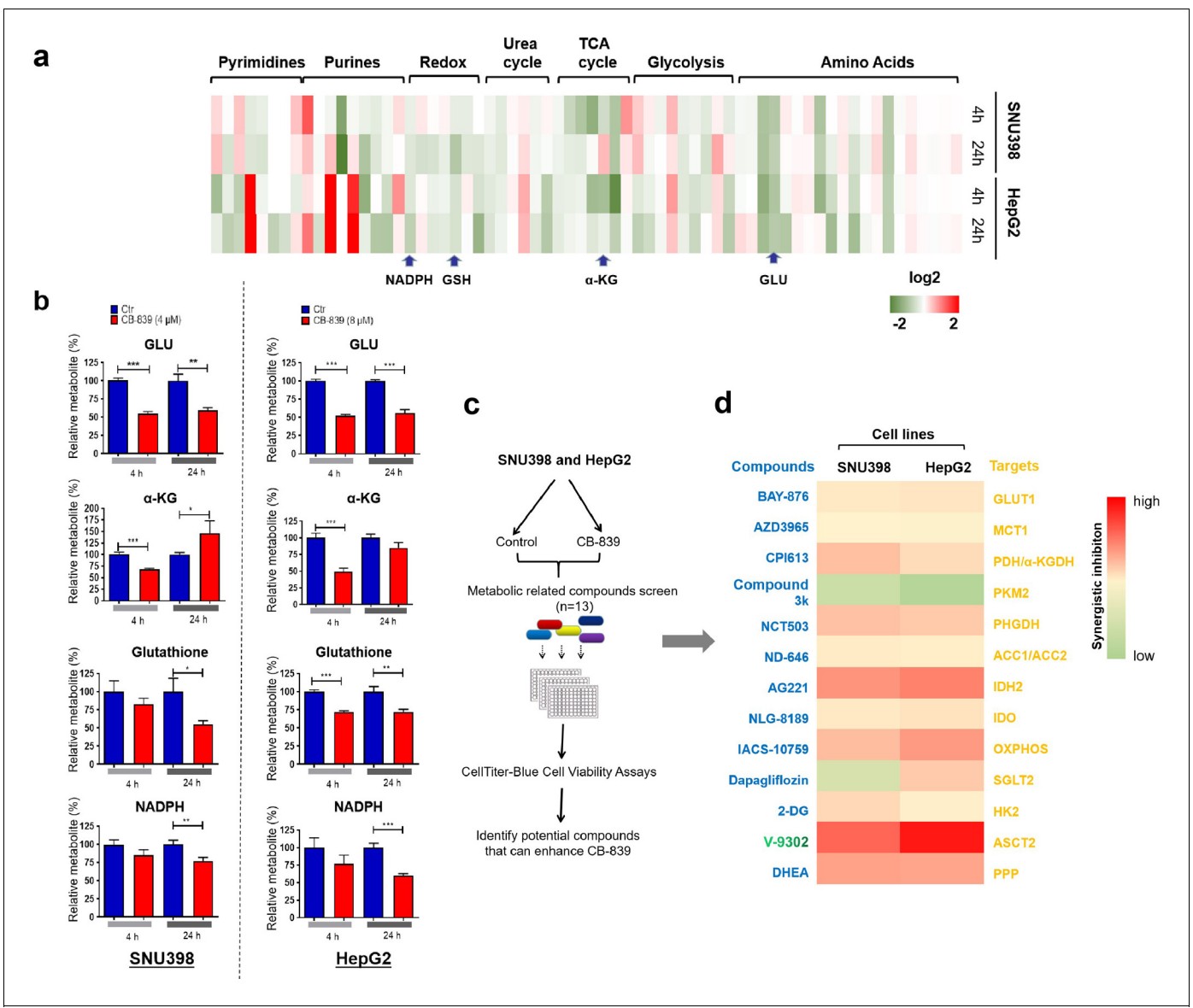

**Figure 3.** A compounds screen identifies ASCT-2 inhibitor V-9302 sensitizing GD liver cancer cells to CB-839 treatment. (**a**) Heatmap representation of 66 metabolites between treated and untreated groups. Intracellular metabolite levels measured by LC/MS-MS in SNU398 and HepG2 cells treated with DMSO or CB-839 (SNU398: 4 μM; HepG2: 8 μM) for 4 and 24 hr, respectively. These metabolites were mapped to seven major pathways including those of the glycolytic system, TCA cycle, urea cycle, redox reaction, purine and pyrimidine metabolism. Each column represented a metabolite. Deeper red color represents higher content; conversely, deeper green color represents lower content. (**b**) Graphic representation of glutamate (GLU), α-ketoglutarate (α-KG), glutathione (GSH), NADPH were shown in the LC/MS-MS screen in **a**. Data are represented as mean ± SEM, n = 3 independent experiments. (**c**) Schematic outline of the compounds screen on the basis of CB-839 treatment: Two GD cell lines, SNU398 and HepG2, were first treated with 4 and 8 μM CB-839, respectively. Then, 13 compounds inhibiting a variety of druggable tumor metabolism targets were tested at their IC50 concentrations for 4 d. (**d**) Heatmap represents the enhanced percentage of viability inhibition by 13 compounds in SNU398 and HepG2, respectively. Deeper red color represents higher enhance; conversely, deeper green color represents lower enhance. Statistical significance was assessed using a Student's *t* test. *p<0.05, **p<0.01, ***p<0.001.

The online version of this article includes the following source data and figure supplement(s) for figure 3:

**Source data 1.** Intracellular metabolite levels were measued by LC/MS-MS in SNU398 and HepG2 cells treated with DMSO or CB-839.
**Source data 2.** Data related to *Figure 3d*.
**Figure supplement 1.** All metabolites shown in *Figure 3a* were labeled.

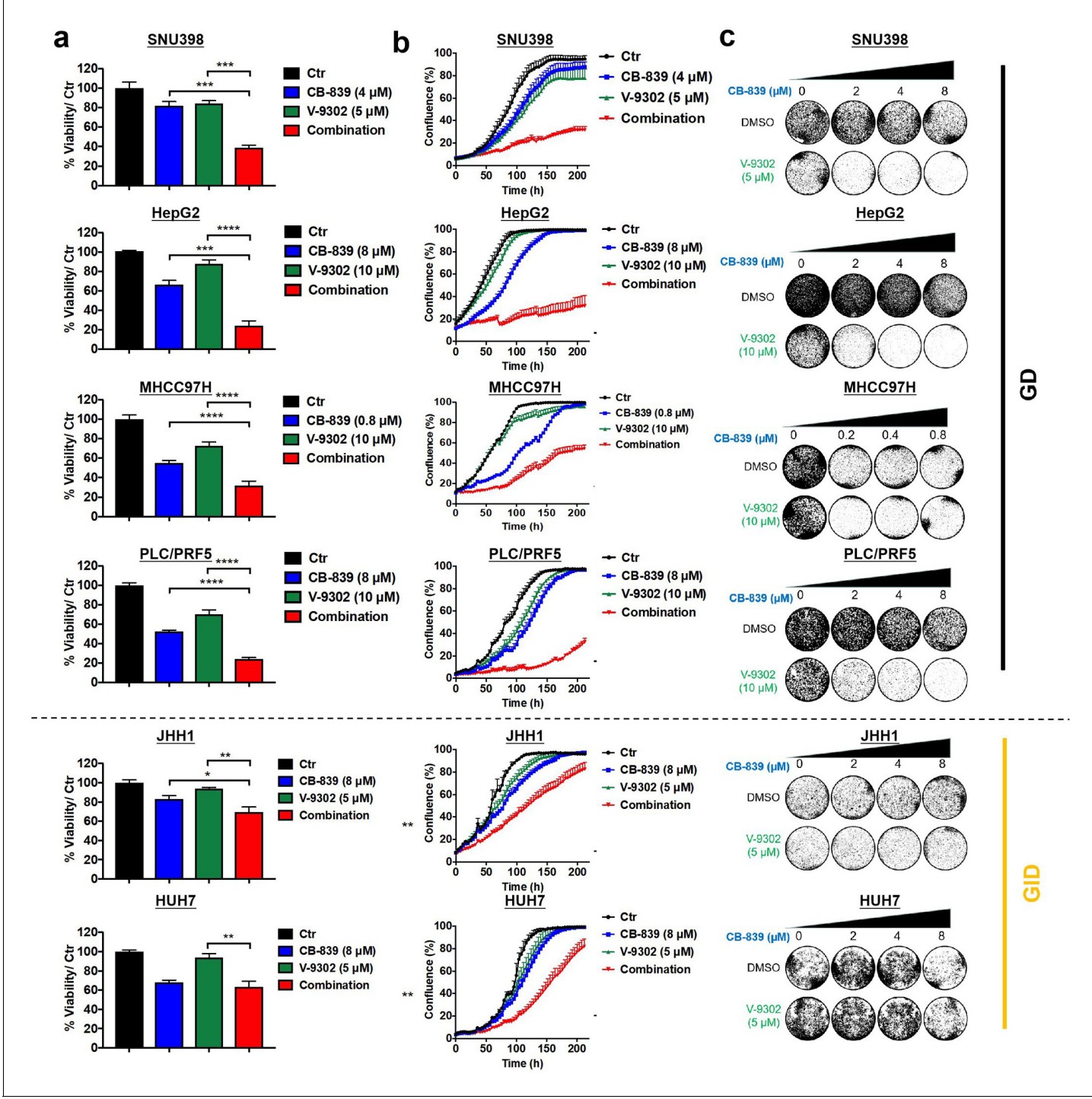

**Figure 4.** Combination of CB-839 and V-9302 shows potential synergy in multiple GD liver cancer cells. (**a-c**) Liver cancer cells (four GD cell lines and two GID cell lines) were treated with CB-839, V-9302, or the combination at the indicated concentration. CellTiter Blue viability assays (**a**), IncuCyte assays (**b**) and long-term colony formation assays (**c**) were performed, respectively. Data are represented as mean ± SEM. Statistical significance was assessed using a Student's $t$ test. *$p<0.05$, **$p<0.01$, ***$p<0.001$.

The online version of this article includes the following source data and figure supplement(s) for figure 4:

**Source data 1.** Combination of CB-839 and V-9302 shows potential synergy in multiple GD liver cancer cells.
**Figure supplement 1.** The combination of CB-839 and V-9302 showed an anti-proliferation effect in GD cell lines in vitro.
**Figure supplement 2.** The combination of BPTES and V-9302 showed an anti-proliferation effect in four GD cell lines, but not in two GID cell lines in vitro.

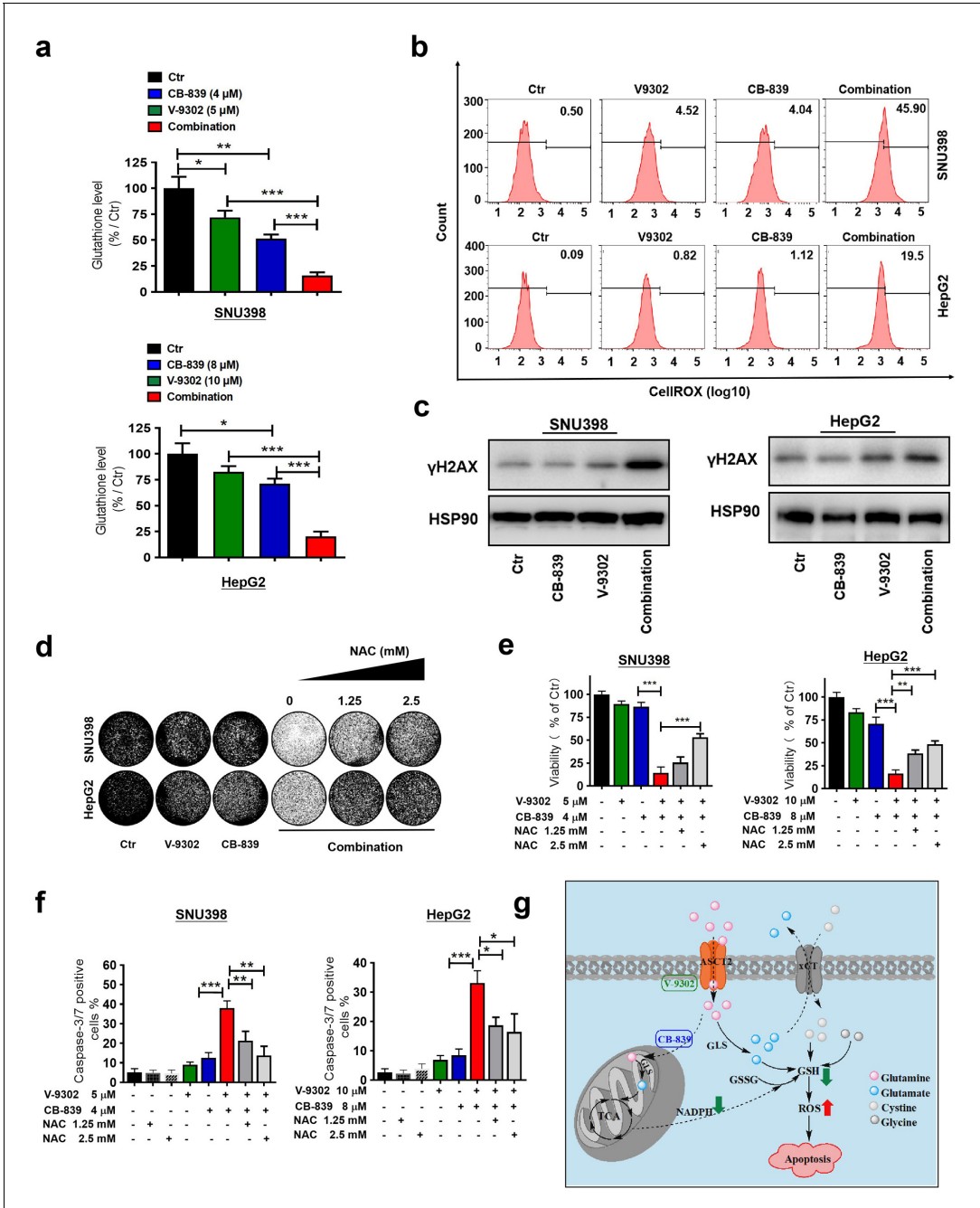

**Figure 5.** Combination of CB-839 and V-9302 depletes GSH and induces lethal ROS levels in GD liver cancer cells. (a) Intracellular GSH levels were measured by LC/MS-MS in SNU398 and HepG2 cells treated with indicated drugs for 48 hr, respectively. (b) ROS levels were measured using the CellROX Deep Red flow cytometry assay. (c) Western blot analysis was performed for γH2AX as a DNA damage marker. HSP90 served as a control. (d, e) Long-term colony formation assay and CellTiter-Blue viability assay show rescued proliferation and viability of SNU398 and HepG2 cells after ROS scavenger N-acetyl-cysteine (NAC) treatment. (f) Caspase-3/7 positive percentages of control, NAC, V-9302, CB-839, the combination, or combination plus NAC treated SNU398 and HepG2 cells in the presence of a caspase-3/7 activatable dye. (g) Schematic showing how the combination of CB-839 and V-9302 decreases GSH and induce apoptosis in liver cancer. All the data in this figure are represented as mean ± SEM. Statistical significance was assessed using a Student's t test. *p<0.05, **p<0.01, ***p<0.001.

The online version of this article includes the following source data for figure 5:

**Source data 1.** Combination of CB-839 and V-9302 depletes GSH and induces lethal ROS level in GD liver cancer cells.

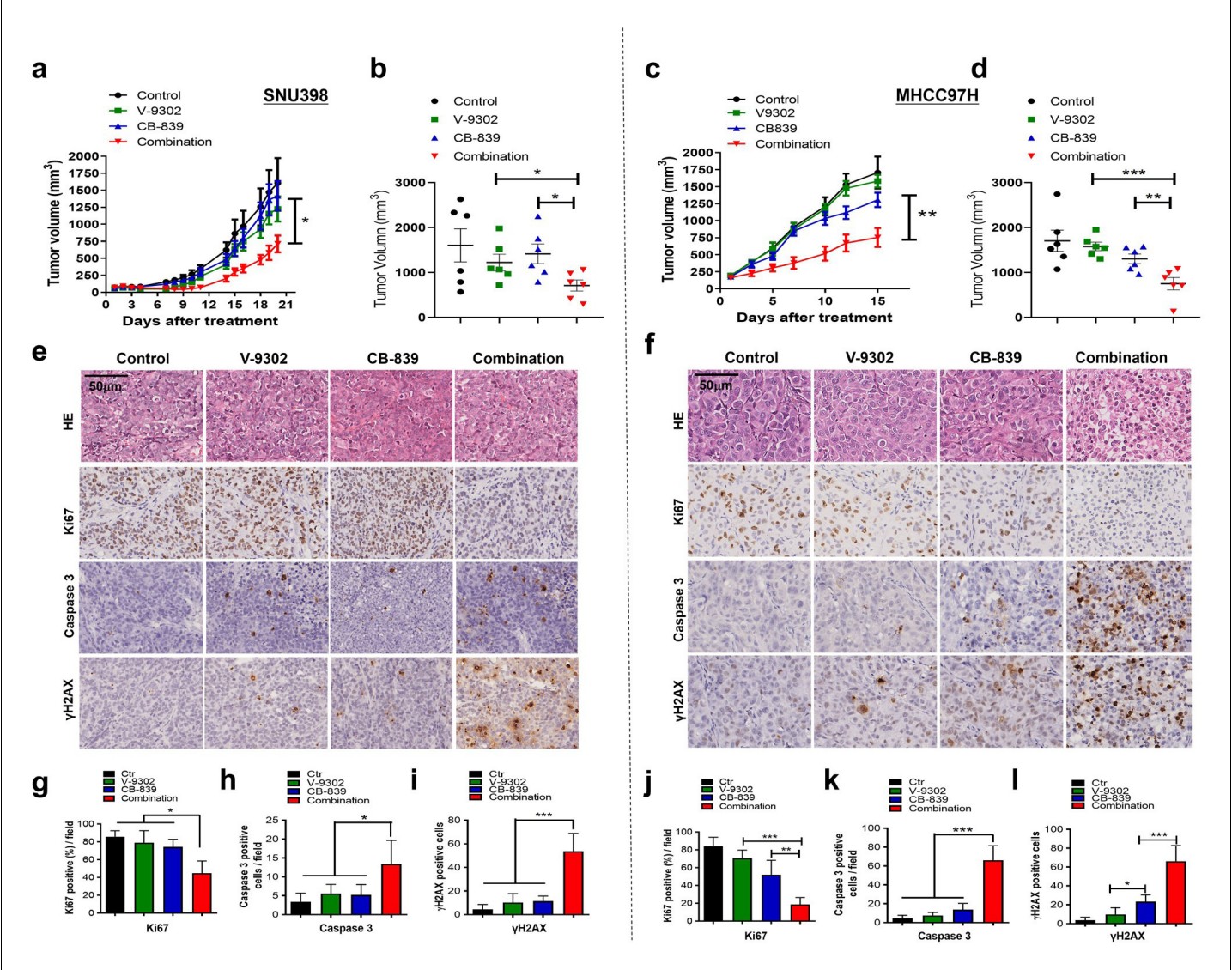

**Figure 6.** Combined treatment inhibits xenograft growth and induces apoptosis in vivo. SNU398 and MHCC97H cells were grown as tumor xenografts in BALB/c nude mice. Longitudinal tumor volume progression in SNU398 and MHCC97H tumor-bearing mice treated with vehicle (n = 6), CB-839 (150 mg/kg, oral gavage, twice per day; n = 6), V-9302 (30 mg/kg, intraperitoneal injection; n = 6), or combined therapies (n = 6) for 20 or 15 d, respectively. Growth curve and endpoint tumor volume of SNU398 (**a, b**) and MHCC97H (**c, d**) xenografts. (**e, f**) Representative images of HE, Ki67, cleaved caspase-3, and γH2AX in SNU398 (**e**) and MHCC97H (**f**) xenograft models. Scale bars = 50 μm. (**g–i**) Quantification of Ki67 positive cells (**g**), cleaved caspase-3 positive cells (**h**), and γH2AX positive cells (**i**) in SNU398 xenografts. (**j–l**) Quantification of Ki67 positive cells (**j**), cleaved caspase-3 positive cells (**k**), and γH2AX positive cells (**l**) in SNU398 xenografts. Data are represented as mean ± SEM. Statistical significance was assessed using a Student's *t* test. *p<0.05, **p<0.01, ***p<0.001.

The online version of this article includes the following source data and figure supplement(s) for figure 6:

**Source data 1.** Combined treatment inhibits xenograft growth and induces apoptosis in vivo.

**Figure supplement 1.** Combination of CB-839 and V-9302 showed no reduction of mice weight in vivo.

## Discussion

Increasing interest in the metabolic vulnerabilities of cancer has given rise to the development of multiple metabolism-targeted therapies targeting diverse aspects of nutrient transport and utilization. For example, de novo pyrimidine synthesis was identified as a metabolic vulnerability of triple-negative breast cancer (TNBC), suggesting that inhibition of pyrimidine synthesis could sensitize TNBC to chemotherapy (*Brown et al., 2017*). Metabolic vulnerabilities of cancer cells can also be

exploited to address drug resistance. Cisplatin-resistant cancer cells were found to be strongly dependent on glutamine for nucleotide biosynthesis and therefore became exquisitely sensitive to treatment with antimetabolites that target nucleoside metabolism (*Obrist et al., 2018*). These findings highlight the possibility of precisely directing the metabolic therapeutics to certain cancer types through identification of their metabolic vulnerabilities.

Liver cancer remains one of the most difficult cancer types to treat due to a paucity of drugs that target critical dependencies and broad-spectrum kinase inhibitors like sorafenib and lenvatinib provide only a modest survival benefit to HCC patients (*Bray et al., 2018*; *Lau, 2008*). Exploiting metabolic vulnerabilities may represent a promising strategy for the treatment of liver cancer. Liver cancer has a metabolic dependency on glutamine. The HGF-MET axis was reported to inhibit pyruvate dehydrogenase complex (PDHC) activity but activate GLS to facilitate glutaminolysis in multiple liver cancer cells (*Huang et al., 2019*). Another study reported that the downregulation of Sirtuin four in liver cancer promoted HCC tumorigenesis via enhancing glutamine metabolism and regulating ADP/AMP levels (*Wang et al., 2019*). In addition, MYC-induced mouse liver tumors had increased GLS expression and decreased glutamate-ammonia ligase (GLUL) expression relative to surrounding tissues, exhibiting elevated Gln catabolism (*Xiang et al., 2015*). Consistently, our study showed that the vast majority of liver cancer cell lines was highly addicted to exogenous Gln in vitro, and the majority of Gln metabolism-related genes was upregulated in tumor tissues of HCC patients. A previous study showed that targeting GLS could attenuate the stemness properties of HCC (*Li et al., 2019*), indicating the possible application of GLS inhibitors in HCC. However, single-drug treatment of CB-839 only showed a very limited anti-tumor effect in most GD liver cancer cells. High-throughput screenings, such as genome-wide CRISPR screen, should be applied to discover the functionally important genes responsible for glutamine dependence or CB-839 sensitivity in liver cancer. The metabolome analysis revealed that CB-839 treatment already caused strong depletion of several key metabolites involved in Gln metabolism (such as GSH), which did not recover even after 24 hr of drug treatment, indicating a possible persistent metabolic vulnerability in the presence of CB-839. This metabolic vulnerability was further exacerbated by ASCT2 inhibitor V-9302, as a result of a further decrease in GSH levels and a lethal increase in the already-elevated levels of ROS. This synergistic effect may be explained by dual inhibition of Gln metabolism. On the one hand, GLS inhibition by CB-839 prevents glutamate production, a direct precursor of GSH synthesis, as well as cystine/glutamate exchange by xCT, leading to a decrease in cystine, another essential precursor for GSH synthesis. On the other hand, pharmacological blockade of the primary transporter of glutamine ASCT2 can decrease the uptake of extracellular glutamine, leading to the shortage of Gln supply and GSH synthesis (*Schulte et al., 2018*). However, there is still some debate about the target selectivity of V-9302 for ASCT2. Broer et al (*Angelika et al., 2018*) reported that V-9302 did not inhibit ASCT2 and glutamine uptake, but rather blocked sodium-neutral amino acid transporter 2 (SNAT2) and the large neutral amino acid transporter 1 (LAT1). Interestingly, it has been reported that, in addition to ASCT2, both SNAT2 and LAT1 also can mediate glutamine uptake in most cancer cells (*Bröer and Bröer, 2017*). Thus, the depression of glutamine uptake by V-9302 remains to be further investigated. Impressively, we find that the CB-839 induced metabolic vulnerability also can be further enhanced by other several anti-metabolic drugs such as OXPHOS inhibitor IACS-10759, PPP inhibitor DHEA, and PHGDH inhibitor NCT503 (*Figure 3d*). This observation suggests the possibility of developing other new combination strategies against liver cancer based on CB-839. However, it is worth noting that our in vitro study was performed over physiological glutamine concentrations, occurring at concentrations that may not be experienced in solid tumors.

The liver clears almost all of the portal vein ammonia, converting it into glutamine and urea, preventing entry into the systemic circulation. Chronic liver disease has impaired liver function and usually leads to dysfunction of ammonia metabolism. For example, there is a general correlation between higher levels of ammonia and more severe encephalopathy in cirrhosis (*Olde Damink et al., 2002*). As a mitochondrial enzyme, GLS plays important roles in liver ammonia metabolism (*Botman et al., 2014*). There are two distinct isoforms of GLS, GLS1 and GLS2, which possess discrete tissue distribution, structural properties, and molecular regulation. The *GLS1* gene encodes two isoforms, kidney-type glutaminase (KGA, long transcript isoform) and the glutaminase C (GAC, short transcript isoform), which are expressed in kidneys and in a variety of other tissues including cancer cells (*Katt et al., 2017*). In our study, we found that the GLS1 expression level was significantly increased in HCC tissues. The *GLS2* gene is strongly expressed and active in periportal

hepatocytes, where they generate glutamate and release ammonia for urea synthesis (*Katt et al., 2017*; *Curthoys and Watford, 1995*). Interestingly enough, CB-839 selectively inhibits GLS1 but not GLS2 (*Gross et al., 2014*). Taken together, although many HCC patients co-exist liver disease and thus usually have impaired ammonia metabolism, GLS inhibition by CB-839 treatment is less likely to pose a problem for these patients.

The advantage of metabolism-targeted therapies is the ability to non-invasively assess the activity of the metabolic pathway in mouse models and even in the clinic. Recently, a voltage-sensitive, positron emission tomography (PET) radiotracer known as [18]F-BnTP was developed to measure mitochondrial membrane potential in non-small-cell lung cancer. It was found that only mouse tumors with a high uptake of [18]F-BnTP showed marked growth suppression when treated with oxidative-phosphorylation inhibitor IACS-010759 (*Momcilovic et al., 2019*). Another example is the widespread use of a radioisotope labeled glucose analog, [18F]fluoro-2-deoxyglucose ([18]F-FDG), in clinical use for diagnosing and staging cancer (*Kelloff et al., 2005*; *Nair et al., 2012*). [18]F-FDG undergoes cellular uptake via the same pathway as glucose but becomes trapped intracellularly after phosphorylation by hexokinase, which can be visualized by PET imaging and used to detect glucose uptake in tumors. In our study, we find that the combination of CB-839 and V-9302 only shows synergy in liver tumors that are addicted to Gln. It suggests that measuring Gln uptake or Gln addiction is of potential importance for selecting patients that might benefit from this combination. One way to accomplish this is to test the Gln dependence of tumor cells in vitro, which might be consistent with their metabolic dependence in vivo. A more accurate and practical approach is developing a Gln-based PET imaging agent. The Gln tracers 5-[11]C-(2S)-glutamine ([11]C-Gln) and [18]F-(2S,4R)4-fluoroglutamine ([18]F-(2S,4R)4-FGln) provide useful tools for probing and monitoring in vivo metabolism of Gln (*Zhu et al., 2017*). Tracer [11]C-Gln was shown to visualize glutaminolytic tumors in vivo as a metabolic marker for probing glutamine-addicted tumor (*Qu et al., 2012*). Similarly, [18]F-(2S,4R)4-FGln PET was reported to track cellular Gln pool size in breast cancers with differential GLS activity and applied as a response marker for CB-839 (*Zhou et al., 2017*). Therefore, the development of isotope-labeled Gln PET imaging may facilitate the translation of the drug combination strategy described here through the identification of those patients that are most likely to benefit.

# Materials and methods

## Key resources table

| Reagent type (species) or resource | Designation | Source or reference | Identifiers | Additional information |
|---|---|---|---|---|
| Cell line (*Homo-sapiens*) | Hep3B | ATCC | Cat#:HB-8064 | |
| Cell line (*Homo-sapiens*) | Huh7 | JCRB | Cat#:JCRB0403; RRID:CVCL_0336 | |
| Cell line (*Homo-sapiens*) | HepG2 | ATCC | Cat# HB-8065; RRID:CVCL_0027 | |
| Cell line (*Homo-sapiens*) | SNU398 | ATCC | Cat# CRL-2233; RRID:CVCL_0077 | |
| Cell line (*Homo-sapiens*) | SNU449 | ATCC | Cat# CRL-2234; RRID:CVCL_0454 | |
| Cell line (*Homo-sapiens*) | Huh6 | RCB | Cat# RCB1367; RRID:CVCL_4381 | |
| Cell line (*Homo-sapiens*) | SK-Hep1 | ATCC | Cat# HTB-52; RRID:CVCL_0525 | |
| Cell line (*Homo-sapiens*) | JHH1 | JCRB | Cat# NIHS0056; RRID:CVCL_2785 | |
| Cell line (*Homo-sapiens*) | SNU387 | ATCC | Cat# CRL-2237; RRID:CVCL_0250 | |
| Cell line (*Homo-sapiens*) | PLC/PRF5 | ATCC | Cat# CRL-802 | |

*Continued on next page*

*Continued*

| Reagent type (species) or resource | Designation | Source or reference | Identifiers | Additional information |
|---|---|---|---|---|
| Cell line (*Homo-sapiens*) | MHCC97H | | RRID:CVCL_4972 | Liver Cancer Institute of Zhongshan Hospital (Shanghai, China) |
| Chemical compound, drug | CB-839 | Selleck Chemicals | S7655 | |
| Chemical compound, drug | BPTES | Selleck Chemicals | S7753 | |
| Chemical compound, drug | BAY-876 | Selleck Chemicals | S8452 | |
| Chemical compound, drug | AZD3965 | Selleck Chemicals | S7339 | |
| Chemical compound, drug | CPI613 | Selleck Chemicals | S2776 | |
| Chemical compound, drug | Compound 3 k | Selleck Chemical | S8616 | |
| Chemical compound, drug | NCT503 | Selleck Chemical | S8619 | |
| Chemical compound, drug | AG221 | Selleck Chemical | S8205 | |
| Chemical compound, drug | NLG-8189 | Selleck Chemical | S7756 | |
| Chemical compound, drug | IACS-10759 | Selleck Chemical | S8731 | |
| Chemical compound, drug | Dapagliflozin | Selleck Chemical | S1548 | |
| Chemical compound, drug | 2-DG | Selleck Chemical | S4701 | |
| Chemical compound, drug | DHEA | Selleck Chemical | S2604 | |
| Chemical compound, drug | ND-646 | MedChemExpress | HY-101842 | |
| Chemical compound, drug | V-9302 | Probechem Biochemicals | 1855871-76-9 | |
| Chemical compound, drug | N-acetyl cysteine (NAC) | Sigma-Aldrich | 616-91-1 | |
| Antibody | Anti-HSP90 (Mouse monoclonal) | Santa Cruz Biotechnology | sc-13119; RRID:AB_675659 | (WB 1:2000) |

*Continued on next page*

*Continued*

| Reagent type (species) or resource | Designation | Source or reference | Identifiers | Additional information |
|---|---|---|---|---|
| Antibody | Anti-GLS (Rabbit polyclonal) | Proteintech | Cat# 12855–1-AP, RRID:AB_2110381 | (WB 1:1000) |
| Antibody | Anti-γH2AX (Rabbit monoclonal) | Cell Signaling Technology | Cat# 9718; RRID:AB_2118009 | (WB 1:1000) (IHC 1:200) |
| Antibody | Anti-Ki67 (Rabbit polyclonal) | Abcam | Cat# ab15580; RRID:AB_443209 | (IHC 1:200) |
| Antibody | Anti-Cleaved Caspase-3 (Rabbit polyclonal) | Abcam | ab2302 | (IHC 1:200) |
| Commercial assay or kit | CellROX Deep Red Flow Cytometry Assay Kit | Life Technologies | C10491 | |
| Software, algorithm | javaGSEA desktop application | http://software.broadinstitute.org/gsea | | |
| Software, algorithm | Prism - Graphpad | https://www.graphpad.com/scientific-software/prism/ | | |

## Cell lines

The human liver cancer cell lines Hep3B, Huh7, HepG2, SNU398, SNU449, Huh6, SK-Hep1, JHH1, SNU387, and PLC/PRF/5 were provided by Erasmus University (Rotterdam, Netherlands). MHCC97H was provided by the Liver Cancer Institute of Zhongshan Hospital (Shanghai, China). The majority of liver cancer cell lines were established from hepatocellular carcinoma (HCC). Among them, SK-Hep1 was established from an endothelial tumor in the liver and Huh6 is a hepatoblastoma cell line. HCC cells were cultured in DMEM with 10% FBS and penicillin/streptomycin (Gibco) at 37°C/5% $CO_2$. All cell lines were tested negative for mycoplasma contamination. The cell lines were authenticated by applying short tandem-repeat (STR) DNA profiling.

## Compounds and antibodies

CB-839 (S7655), BPTES (S7753), BAY-876 (S8452), AZD3965 (S7339), CPI613 (S2776), Compound 3 k (S8616), NCT503 (S8619), AG221 (S8205), NLG-8189 (S7756), IACS-10759 (S8731), Dapagliflozin (S1548), 2-DG (S4701) and DHEA (S2604) were purchased from Selleck Chemicals. ND-646 (HY-101842) was purchased from MedChemExpress. V-9302 (1855871-76-9) was purchased from Probechem Biochemicals. N-acetyl cysteine (NAC) was purchased from Sigma. Antibody against HSP90 (sc-13119) was purchased from Santa Cruz Biotechnology. Antibody against two different splice forms of GLS, KGA/GAC, (12855–1-AP) was purchased from Proteintech. Antibody against γH2AX (#9718) was purchased from Cell Signaling. Antibodies against Ki67 (ab15580) and Cleaved caspase-3 (ab2303) were from Abcam.

## Protein lysate preparation and immunoblotting

Cells were washed with PBS and lysed with RIPA buffer supplemented with Complete Protease Inhibitor (Roche) and Phosphatase Inhibitor Cocktails II and III (Sigma). Protein quantification was performed with the BCA Protein Assay Kit (Pierce). All lysates were freshly prepared and processed with Novex NuPAGE Gel Electrophoresis Systems (Thermo Fisher Scientific) followed by western blotting.

## Long-term colony formation assays

Cells were cultured and seeded onto 6-well plates at a density of $2–10 \times 10^4$ cells per well, depending on the growth rate, and were cultured in normal DMEM medium containing 4 mM glutamine (11995073, ThermoFisher), DMEM medium without glutamine (10313021, ThermoFisher), or the indicated drugs for 10–14 d (medium was changed twice a week). Cells were then fixed with 4% formaldehyde in PBS and stained with 0.1% crystal violet diluted in water.

## Incucyte cell proliferation assay and apoptosis assay

Indicated cell lines were seeded onto 96-well plates at a density of 1000–8000 cells per well, depending on the growth rate and design of the experiments. About 12 hr after seeding, cells were cultured in medium with drugs of indicated concentrations using the HP D300 Digital Dispenser (HP) and imaged every 4 hr in Incucyte ZOOM (Essen Bioscience). Phase-contrast images were analyzed to detect cell proliferation based on cell confluence. For cell apoptosis, caspase-3/7 green apoptosis assay reagent was added to the culture medium and cell apoptosis was analyzed based on green fluorescent staining of apoptotic cells.

## CellTiter blue viability assays

Cell lines were cultured and seeded into 96-well plates (2000–5000 cells per well). After about 12 hr after seeding, drugs with the indicated concentrations were added to liver cancer cells. Cell viability was measured with the CellTiter-Blue assay (Roche) after treatment with the drug for 72 hr. The relative viability of different cell lines in the presence of drug was normalized against control conditions (untreated cells) after subtraction of the background signal.

## ROS detection

The cells were treated in the absence or presence of drugs for 48 hr. ROS level in cells was detected using CellROX Deep Red Flow Cytometry Assay Kit (C10491, Life Technologies) according to the manufacturer's instructions.

## Immunohistochemical staining and scoring

HCC specimens were obtained from patients who underwent curative surgery in Eastern Hepatobiliary Hospital of the Second Military Medical University in Shanghai, China. Patients were not subjected to any preoperative anticancer treatment. Ethical approval was obtained from the Eastern Hepatobiliary Hospital Research Ethics Committee and written informed consent was obtained from each patient. Immunohistochemistry (IHC) was performed according to our previous study (*Jin et al., 2017*). Briefly, formalin-fixed paraffin-embedded samples from HCC patients were probed with the GLS1 antibody (12855–1-AP, Proteintech). Formalin-fixed paraffin-embedded samples were also obtained from xenograft tumors and probed with antibodies against Ki-67 (sc-23900, Santa Cruz), against γH2AX (#9718) and cleaved caspase-3 (ab2303, Abcam). Following incubation with the primary antibodies, positive cells were visualized using DAB+ as a chromogen.

Semiquantitative scores were used to analyze the immunostaining of each HCC case in tissue microarray. Intensity score of staining was categorized into 0 (-), 1 (+), 2 (++), or 3 (+++), denoting negative, weak, moderate, or strong staining, respectively. Percentage score of immunostaining was categorized into 0 (0–5%), 1 (6–25%), 2 (26–50%), 3 (51–75%), or 4 (>76%) based on the percentage of positive cells. Three random microscope fields per tissue were calculated. The sum of intensity and percentage of staining was used as the final score of expression level and determined by the formula: final score = intensity score $\times$ percentage score. The final score of $\leq 4$ was defined as a low expression of GLS1 and >4 as a high expression of GLS1.

## Metabolomics

Cells were cultured in 6-well plates until 60% confluent. The medium was replaced 24 hr before harvesting. Then, cells were treated with DMSO and CB-839 (4 μM for SNU398 and 8 μM for HepG2) for 4 hr and 24 hr, respectively. After washing with ice-cold PBS, metabolites were extracted from cells in 0.5 mL lysis buffer containing methanol/acetonitrile/dH$_2$O (2:2:1). Samples were spun at 16,000 $\times$ g for 15 min at 4°C. Supernatants were collected for LC-MS analysis.

LC-MS analysis was performed on an Exactive mass spectrometer (Thermo Fisher Scientific) coupled to a Dionex Ultimate 3000 autosampler and pump (Thermo Fisher Scientific). The MS operated in polarity-switching mode with spray voltages of 4.5 and −3.5 kV. Metabolites were separated using a SeQuant ZIC-pHILIC HPLC Columns (2.1 mm × 150 mm, 5 µm, guard column 2.1 mm × 20 mm, 5 µm; Merck) using a linear gradient of acetonitrile and eluent A [20 mM $(NH_4)_2CO_3$, 0.1% $NH_4OH$ in ULC/MS grade water (Biosolve)]. The flow rate was set at 150 µL/min. Metabolites were identified and quantified using LCQUANTM Quantitative Software (Thermo Fisher Scientific) on the basis of exact mass within 5 ppm and further validated by concordance with retention times of standards. Metabolites were quantified using LCQUANTM Quantitative Software (Thermo Fisher Scientific). Peak intensities were normalized based on median peak intensity.

## Compound screen

SNU398 and HepG2 cells were seeded in 96-well plates, respectively. A total of 13 compounds inhibiting a variety of druggable tumor metabolism targets were independently added into the plates at a certain concentration gradient, and cultured for 4 d. Then IC50 concentrations for each compound were analyzed. Then, SNU398 and HepG2 cells were firstly treated with 4 and 8 µM CB-839, respectively, and further treated with IC50 concentrations of each compound. For the compound has IC50 $\geq$100 µM, the concentration of 100 µM was used. For 2-DG, a routine concentration at the mM level was used. The synergistic viability inhibition was analyzed using the following formula: synergistic inhibition = inhibition of combination − inhibition of CB-839 × (1 + inhibition of candidate compound).

## Xenografts model

All animals were manipulated according to protocols approved by the Shanghai Medical Experimental Animal Care Commission and the Shanghai Cancer Institute. SNU398 cells (8 × $10^6$ cells per mouse) and MHCC97H (6 × $10^6$ cells per mouse) were injected subcutaneously into the right posterior flanks of 6-week-old BALB/c nude mice (six mice per group), respectively. Tumor volume based on caliper measurements was calculated by the modified ellipsoidal formula: tumor volume = ½ length × width. When tumors reached a volume of approximately 50–100 mm³, mice were randomly assigned to 5 d/week treatment with vehicle, CB-839 (150 mg/kg, oral gavage, twice per day), V-9302 (30 mg/kg, intraperitoneal injection), or a drug combination in which each compound was administered at the same dose and scheduled as single agents.

## Statistics

Statistical significance was calculated by Student's $t$ test with two tails. All data are expressed as mean ± SEM. Prism and Microsoft Excel were used to generate graphs and statistical analyses. *p value < 0.05, **p value < 0.01, ***p value < 0.001.

## Acknowledgements

This work was supported by grants from the Dutch Cancer Society (KWF) through the Oncode Institute, the National Natural Science Foundation of China (81702838, 81920108025), National Science and Technology Key Project of China (2018ZX10302205), Shanghai Rising-Star Program (19QA1408200). Shanghai Municipal Commission of Health and Family Planning (2018YQ20).

## Additional information

### Funding

| Funder | Grant reference number | Author |
| --- | --- | --- |
| Dutch Cancer Society | KWF | Rene Bernards |
| National Science and Technology Key Project of China | 2018ZX10302205 | Haojie Jin |
| National Natural Science Foundation of China | 81702838 | Haojie Jin |

| National Natural Science Foundation of China | 81920108025 | Wenxin Qin |
|---|---|---|
| Shanghai Rising-Star Program | 19QA1408200 | Haojie Jin |
| Shanghai Municipal Commission of Health and Family Planning | 2018YQ20 | Haojie Jin |

The funders had no role in study design, data collection and interpretation, or the decision to submit the work for publication.

## Author contributions

Haojie Jin, Conceptualization, Data curation, Funding acquisition, Investigation, Writing - original draft, Project administration, Writing - review and editing; Siying Wang, Data curation, Validation, Investigation; Esther A Zaal, Data curation, Software, Investigation, Methodology; Cun Wang, Suggestion; Haiqiu Wu, Astrid Bosma, Data curation, Investigation; Fleur Jochems, Investigation, Suggestion; Nikita Isima, Resources, Data curation, Validation, Investigation; Guangzhi Jin, Resources, Data curation, Software, Formal analysis, Methodology; Cor Lieftink, Data curation, Software, Formal analysis, Validation, Methodology; Roderick Beijersbergen, Conceptualization, Supervision, Writing - review and editing, Suggestion; Celia R Berkers, Wenxin Qin, Rene Bernards, Conceptualization, Supervision, Funding acquisition, Project administration, Writing - review and editing

## Author ORCIDs

Haojie Jin https://orcid.org/0000-0001-9295-1951
Rene Bernards https://orcid.org/0000-0001-8677-3423

## Ethics

Human subjects: Ethical approval was obtained from the Eastern Hepatobiliary Hospital Research Ethics Committee (EHBHKY2014-03-006), and written informed consent was obtained from each patient.
Animal experimentation: All animals were manipulated according to protocols approved by the Shanghai Medical Experimental Animal Care Commission and Shanghai Cancer Institute.

## Decision letter and Author response

Decision letter https://doi.org/10.7554/eLife.56749.sa1
Author response https://doi.org/10.7554/eLife.56749.sa2

# Additional files

## Supplementary files
• Transparent reporting form

## Data availability

All data generated or analysed during this study are included in the manuscript and supporting files.

The following previously published datasets were used:

| Author(s) | Year | Dataset title | Dataset URL | Database and Identifier |
|---|---|---|---|---|
| The Cancer Genome Atlas Research Network | 2017 | Liver Hepatocellular Carcinoma | https://portal.gdc.cancer.gov/projects/TCGA-LIHC | The Cancer Genome Atlas, TCGA-LIHC |
| The Cancer Genome Atlas Research Network | 2017 | Cholangiocarcinoma | https://portal.gdc.cancer.gov/projects/TCGA-CHOL | The Cancer Genome Atlas, TCGA-CHOL |
| Wang XW | 2010 | Gene expression data of human hepatocellular carcinoma (HCC) | https://www.ncbi.nlm.nih.gov/geo/query/acc.cgi?acc=GSE14520 | NCBI Gene Expression Omnibus, GSE14520 |

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
