## [Decision Letter]

**Acceptance summary:**

The authors report that glutaminase inhibitors synergize with inhibitors of the neutral amino acid transporter ASCT2 as a potential treatment for hepatocellular carcinoma. They show that small molecule inhibition of ASCT2 synergizes with glutaminase inhibitors in cell lines that do not respond to glutaminase inhibitor alone. They also show that the combination drug treatment decreases glutathione levels and increases intracellular ROS, suggesting that glutathione depletion is a liability of glutamine starvation in liver cancer cells. The drug combination also slows tumor growth in xenograft models suggesting this drug combination warrants further study as a treatment for hepatocellular carcinoma.

**Decision letter after peer review:**

Thank you for submitting your article "A powerful drug combination strategy targeting glutamine addiction for the treatment of liver cancer" for consideration by *eLife*. Your article has been reviewed by two peer reviewers, including Matthew G Vander Heiden as the Reviewing Editor and Reviewer #1, and the evaluation has been overseen by Jonathan Cooper as the Senior Editor.

The reviewers have discussed the reviews with one another and the Reviewing Editor has drafted this decision to help you prepare a revised submission.

Summary:

The authors report that glutaminase inhibitors synergize with inhibitors of the neutral amino acid transporter ASCT2 as a potential treatment for hepatocellular carcinoma. They show that small molecule inhibition of ASCT2 synergizes with glutaminase inhibitors in cell lines that do not respond to glutaminase inhibitor alone. They also show that the combination drug treatment decreases glutathione levels and increases intracellular ROS, and treatment with the antioxidant N-acetylcysteine provides a partial rescue, suggesting that glutathione depletion is a liability of glutamine starvation in liver cancer cells. Finally, they observe the drug combination results in ~40% lower tumor volume and increased expression of markers of DNA damage and apoptosis in xenograft models.

Essential revisions:

Please see comments from both reviewers below. Please take all suggestions into consideration, but the following should be addressed with either data or discussion in a revised manuscript if possible:

1) Is any information available on how responder cells differ from non-responder cells? If clues can be gleaned from the Cancer Cell Line Encyclopedia and the Dependency Map (depmap.org) based on mutational status or gene expression please discuss this in the paper. Even in the absence of this, considering whether β-catenin mutations correlates with response should be considered per reviewer 2.

2) If possible please address how much V-9302 affects amino acid uptake from the media. If this is not possible, a discussion of the Broer et al. Front Pharmacol 2018 paper suggesting V-9302 is not selective for ASCT2 is warranted.

3) Please clarify how the 4 GD cell lines were chosen to test the drug combination chosen and provide data for other cells, including the GID lines, if available.

4) Please comment on expression of the ASCT1 splice variants, and consider whether this tracks with differential sensitivity (see reviewer 2 comment 1).

5) Please discuss potential issues of targeting glutamine metabolism in patients with HCC who also have impaired ammonia metabolism. If possible test ammonia levels in mice treated with the drug combination.

6) Please clarify how the 13 different "metabolic drugs" were selected to test for synergy with CB-839.

7) Please reconsider interpretation of Figure 5 per reviewer 2 point 5, as the effects on glutathione seems additive, while the effects on ROS and gH2AX are clearly synergistic. This indicates that there could be other reasons for the ROS, gH2AX effects.

8) Reviewer 2 noted that the NAC effect in Figure 5F is not dramatic, and not dose dependent in HEPG2 cells. If feasible, please repeat this experiment and test the effect of NAC alone to compare with the combination affect.

9) Please address additional points of clarification raised by the reviewers:

– All metabolites shown in Figure 3A should be more clearly labeled and/or presented in a supplementary figure.

– Make clear that KGA/GAC represent different splice forms of GLS.

– Discuss levels of glutamine in media relative to those in blood.

– Clarify if Figure 3B is a graphic representation of the findings of the MS screen (shown in Figure 3A), or results of an independent set of experiments for verification.

– Define that xCT = cystine-glutamate antiporter system for a broad audience.

Full reviewer comments for your reference:

Reviewer #1:

Jin et al. report that the glutaminase inhibitor CB-839 synergizes with the putative ASCT2 inhibitor V-9302 in models of hepatocellular carcinoma. They first confirm a previously observed dependency on glutamine in the media for most liver cancer cell lines to grow in culture, and note that this does not translate into sensitivity to the glutaminase inhibitor CB-839. Through a small molecule screen, they find that V-9302, thought to inhibit the neutral amino acid transporter ASCT2, synergizes with CB-839 in four of the seven glutamine-dependent models that did not respond strongly to CB-839. They show that the combination drug treatment decreases glutathione levels and increases intracellular ROS, and treatment with the antioxidant N-acetylcysteine provides a partial rescue, suggesting that glutathione depletion is a major liability of glutamine starvation in liver cancer cells. Finally, they observe the drug combination results in ~40% lower tumor volume and increased expression of markers of DNA damage and apoptosis in xenograft models.

The result that liver cancer cells respond heterogeneously to glutaminase inhibition is interesting, and fits with glutamine having other fates, including glutathione, that are independent of glutaminase. This is a nice point as many equate glutaminase and glutamine dependency, and the point could be better discussed. Some additional suggestions include:

1) A major questions raised by this work is what underlies such differential sensitivity to targeting of glutamine metabolism across the cells considered. While they understandably focus on the responders, is there any clue as to how they differ from the non-responders? Perhaps there are clues from the Cancer Cell Line Encyclopedia and the Dependency Map (depmap.org), or based on mutational status or gene expression?

2) Recent work (Broer et al. Front Pharmacol 2018) has suggested that V-9302 is not selective for ASCT2 and instead binds other amino acid transporters such as SNAT2 and LAT1, and that V-9302 could be acting by inhibiting uptake of other amino acids. Are data available as to how much V-9302 reduces glutamine uptake from media?

3) It would be useful as a resource for the field if all metabolites shown in Figure 3A were labeled and/or presented in a supplementary figure.

4) As a minor point, suggesting CB-839 is increasingly used to treat cancer in combination is potentially misleading. It is being tested in many trials, but is not yet proven to be effective consistently in any cancer.

5) How were the 4 GD cell lines to test the combination chosen? If data are available for other cells, including the GID lines, that information should be included.

6) The authors might consider that invoking TCA-cycle dependent NADPH production as an explanation for why ROS is increased with the drug combination is potentially inconsistent with the NAC rescue. NADPH is also required to keep NAC reduced in cells similarly to GSH.

Reviewer #2:

Following analysis of glutamine metabolism in HCC cell lines, and bioinformatics analysis of human HCC Haojie Jin et al. revealed that combined inhibition of mitochondrial glutaminase (GLS) and the glutamine transporter ASCT2, potently inhibits growth of many human HCC cell lines. Two different inhibitors of ASCT2 were similarly potent in combination with CB-839 – a GLS inhibitor. The combination increased depleted glutathione levels, increased ROS levels, and increased DNA damage marker levels and apoptosis. The latter were dependent on ROS as NAC treatment abolished the DDR and apoptosis. The same combination was tested in vivo, with seemingly good tolerability, and resulted in a twofold reduction in tumor volume, which was much better than each of the drugs alone.

Tumor metabolism is gaining interest lending hope that metabolic interventions will move into the treatment arena. Parenthetically, they are already there for many years as many of the classic chemotherapy drugs are "metabolic" molecules, interfering with nucleoside metabolism, lending hope that this is indeed a productive path to follow. HCC incidence is rapidly rising, and there are so far no treatments that really succeed in reducing mortality. Thus, the manuscript brings together two highly important topics and reveal a novel metabolic vulnerability of HCC.

1) Glutamine synthetase is overexpressed in HCC, and catalyzes the reverse reaction from GLS, so there should be sufficient levels of cytoplasmic glutamine. ASCT1 is expressed on both the cell membrane and mitochondrial membrane, using two different splice variants (https://doi.org/10.1016/j.cmet.2019.11.020), which can be discriminated by Western blot. I wonder whether the differential sensitivity of the different cell lines relates to the relative abundance of the two ASCT1 variants. This could strengthen the hypothesis that the drug combination targets mitochondrial glutamine metabolism.

2) GLS1 is a mitochondrial enzyme which is predominantly expressed at the periportal zone of the liver acinus. It plays key roles in ammonia metabolism. This could pose a problem for patients with HCC, in many of which ammonia metabolism is already impaired due to co-existing liver disease. The authors should at least discuss this issue, or possibly test ammonia levels in mice treated with the different drug combinations.

3) β–catenin is mutated in a significant number of human HCCs and directly regulates hepatocyte glutamine metabolism. Could the authors correlate β-catenin mutation in the cell line and human cohorts with CB-839 response and glutamine metabolism respectively?

4) The authors tested whether 13 different "metabolic drugs" synergized with CB-839. Can they elaborate on how these were selected?

5) In Figure 5 the effects on glutathione seems additive, while the effects on ROS and gH2AX are clearly synergistic. This indicates that there could be other reasons for the ROS, gH2AX effects. While NAC treatment indicates that ROS are causal in gH2AX and apoptosis, it doesn't directly implicate glutathione depletion as the cause for increased ROS. The interpretation of the data should be changed accordingly.

---

## [Author Response]

Essential revisions:Please see comments from both reviewers below. Please take all suggestions into consideration, but the following should be addressed with either data or discussion in a revised manuscript if possible:1) Is any information available on how responder cells differ from non-responder cells? If clues can be gleaned from the Cancer Cell Line Encyclopedia and the Dependency Map (depmap.org) based on mutational status or gene expression please discuss this in the paper. Even in the absence of this, considering whether β-catenin mutations correlates with response should be considered per reviewer 2.

Following the reviewers’ suggestion, we tried to analyze the data from Cancer Cell Line Encyclopedia and the Dependency Map. However, we found several cell lines used in our study, such as Hep3B and MHCC97H, are not included in these databases. Additionally, we also checked both the presence of mutations in β-catenin and the relative activity of β-catenin of liver cancer cell lines as described in previous publications [1,2]. As shown in Author response table 1 below, there is no obvious correlation between the mutation of β-catenin nor the activity β-catenin and glutamine dependency in our panel of cell lines.

To study the underling mechanism that might explain why different liver cancer cell lines have different dependence on glutamine, we have recently performed a genome-wide CRIPSPR resistance screen in a GD cell line SNU398 in the absence of glutamine. We have identified several candidate genes that can make GD cells resistant to deprivation of exogenous glutamine or the drug combination, but it is too early to include these tentative results (that currently also lack a mechanistic basis) in the current manuscript. According to the policy of *eLife*, we plan to report the data in a preprint on bioRxiv or medRxiv in the future, which would be linked to the original paper. Accordingly, we have discussed this approach in our revised manuscript (Discussion). Because the data shown in Author response table 1 are not informative, we decided to not include it in the manuscript.

Author response table 1

Mutation or relative activity of β-catenin in HCC cell lines. Cell lines underlined are glutamine independent (GID) cell lines.

2) If possible please address how much V-9302 affects amino acid uptake from the media. If this is not possible, a discussion of the Broer et al. Front Pharmacol 2018 paper suggesting V-9302 is not selective for ASCT2 is warranted.

We regret that we currently cannot provide data about how much V-9302 affects amino acid uptake from the media. However, according to the reviewer’s suggestion, we have discussed the Broer et al. Front Pharmacol 2018 paper in our revised manuscript as follows:

“However, there is still some debate about the target selectivity of V-9302 for ASCT2. Broer et al.[3] reported that V-9302 does not inhibit ASCT2 and glutamine uptake, but rather blocked sodium-neutral amino acid transporter 2 (SNAT2) and the large neutral amino acid transporter 1 (LAT1). […] Thus, the depression of glutamine uptake by V-9302 remains to be further investigated.”

Accordingly, we have added it into our revised manuscript (Discussion).

3) Please clarify how the 4 GD cell lines were chosen to test the drug combination chosen and provide data for other cells, including the GID lines, if available.

To test the synergistic anti-proliferation effect of CB-839 and V-9302, we tested 4 GD cell lines, which are not sensitive to single CB-839. Following the reviewer’s suggestion, we further tested the drug combination in other GD and GID cell lines by using colony formation assays. These data are shown in the revised manuscript (Figure 4—figure supplement 1) and mentioned the result in the main text (subsection “A compounds screen identifies that ASCT-2 inhibitor V-9302 sensitizes GD liver cancer cells to CB-839 treatment”).

In short, the combination only showed synergistic anti-proliferation effect in GD cell lines, but only showed very limited effect in two GID cell lines in vitro.

4) Please comment on expression of the ASCT1 splice variants, and consider whether this tracks with differential sensitivity (see reviewer 2 comment 1).

According to the reviewer’s suggestion, we analyzed the two different splice variants of ASCT2 by Western blot. As shown bin Author response image 1, we found no correlation between ASCT2 or mitochondrial ASCT2 and drug sensitivity in our panel of liver cancer cell lines. We decided to not include these data in the revised manuscript, as they do not provide important new insights. However, as mentioned in question 1, we have performed a genome-wide resistant screen with or without glutamine in a GD cell line and identified several candidate genes which may explain the varying sensitivity of the cell lines to the drug combination or exogenous glutamine. According to the policy of *eLife*, we plan to report the data in a preprint on bioRxiv or medRxiv in the future, which would be linked to the original paper.

**Author response image 1. respfig1:** Western blot analysis of the two different splice variants of ASCT2 in liver cancer cell lines.

5) Please discuss potential issues of targeting glutamine metabolism in patients with HCC who also have impaired ammonia metabolism. If possible test ammonia levels in mice treated with the drug combination.

Unfortunately, we don’t have fresh samples from in vivo experiments for ammonia analysis to answer the reviewer’s question. For as far as we know, ammonia measurements of tissues and plasma are problematic due to the volatile nature of ammonia, resulting potentially in false positive or false negative readings. Instead, we discussed this issue in our revised manuscript as follows:

“The liver clears almost all of the portal vein ammonia, converting it into glutamine and urea, preventing entry into the systemic circulation. […] CB-839 selectively inhibits GLS1 but not GLS2 [Gross et al., 2014]. Taken together, although many HCC patients have co-existing liver disease and thus usually have impaired ammonia metabolism, GLS inhibition by CB-839 treatment is less likely to pose a problem for these patients.”

We have added the discussion in the revised manuscript (Discussion).

6) Please clarify how the 13 different "metabolic drugs" were selected to test for synergy with CB-839.

To further exploit the metabolic vulnerability caused by CB-839, we selected metabolism-related drugs (13 compounds) which act in the major metabolic pathways including glycolysis, Krebs cycle, oxidative phosphorylation, pentose phosphate pathway, fatty acid β-oxidation, and gluconeogenesis. We have listed the detail targets of the 13 compounds in the Figure 3—source data 2.

7) Please reconsider interpretation of Figure 5 per reviewer 2 point 5, as the effects on glutathione seems additive, while the effects on ROS and gH2AX are clearly synergistic. This indicates that there could be other reasons for the ROS, gH2AX effects.

In our first submission, we analyzed the ROS level and γH2AX expression after 48 hours of drug treatment. However, we tested the glutathione levels after 4 and 24 hours of drug treatment, respectively. To further confirm the synergistic effect on glutathione levels, we treated the SNU398 cells and HepG2 cells with CB-839, V-9302, or their combination for 48 hours and measured the intracellular glutathione levels. These new data are now shown in Figure 5A of the revised manuscript. In short, we observe that the combination of CB-839 and V-9302 caused an obvious synergistic reduction in glutathione levels after 48 hours of drug treatment.

8) Reviewer 2 noted that the NAC effect in Figure 5F is not dramatic, and not dose dependent in HEPG2 cells. If feasible, please repeat this experiment and test the effect of NAC alone to compare with the combination affect.

According to the reviewers’ suggestion, we have repeated the experiment of HepG2 cells of Figure 5F. As shown in the revised Figure 5F, the two concentrations of NAC (1.25 mM and 2.5 mM) have no effect on apoptosis of HepG2 cells, while both concentrations can rescue combination-induced apoptosis. However, we still cannot observe clear dose dependent rescue, which suggests that the lower concentration of NAC (1.25 mM) is sufficient to achieve maximal effect in HepG2 cells.

9) Please address additional points of clarification raised by the reviewers:– All metabolites shown in Figure 3A should be more clearly labeled and/or presented in a supplementary figure.

According to the reviewer’s suggestion, all metabolites shown in Figure 3A are now presented in Figure 3—figure supplement 1 and we also provide the related source file.

– Make clear that KGA/GAC represent different splice forms of GLS.

We have stated that KGA/GAC represent two different splice forms of GLS in the revised manuscript (Discussion and subsection “Compounds and antibodies”).

– Discuss levels of glutamine in media relative to those in blood.

As the reviewers mentioned, the normal concentration of glutamine in human plasma is usually around 0.6-0.9mM[1]. In tissues, such as the liver and the skeletal muscles, glutamine concentration is even higher than in plasma[4]. Considering the fact that glutamine is easily broken down in vitro, it is difficult to mimic the glutamine concentration of the tumor microenvironment in 2D culture models over time. To maintain the best viability of tumor cells in vitro, we performed all the experiments in normal DMEM medium with sufficient glutamine (4 mM). Similarly, other studies [Schulte et al., 2018; 5] also used normal culture medium with sufficient glutamine to test the drugs response on tumor cells in vitro. Accordingly, we have described it in the Discussion of our revised manuscript.

– Clarify if Figure 3B is a graphic representation of the findings of the MS screen (shown in Figure 3A), or results of an independent set of experiments for verification.

We have stated that Figure 3B is a graphic representation of the findings of the MS screen in our revised manuscript (Figure 3 legend).

– Define that xCT = cystine-glutamate antiporter system for a broad audience.

We have defined xCT with cystine-glutamate antiporter system in our revised manuscript (subsection “Combination of CB-839 and V-9302 depletes glutathione and induces lethal ROS level in GD liver cancer cells”).

In addition, we found SNU182 cells, which were used in Figure 1A and B of our first submission, had mycoplasma contamination. Therefore, we removed the related data of SNU182 cells in Figure 1A and B of our revised Figure 1. We are sorry for this mistake.

References:

1) Ding Z, Shi C, Jiang L, Tolstykh T, Cao H, Bangari DS, Ryan S, Levit M, Jin T, Mamaat K, Yu Q, Qu H, Hopke J, Cindhuchao M, Hoffmann D, Sun F, Helms MW, Jahn-Hofmann K, Scheidler S, Schweizer L, Fang DD, Pollard J, Winter C, Wiederschain D. Oncogenic dependency on β-catenin in liver cancer cell lines correlates with pathway activation. Oncotarget. 2017 Sep 28;8(70):114526-114539

2) Wang W, Xu L, Liu P, Jairam K, Yin Y, Chen K, Sprengers D, Peppelenbosch MP, Pan Q, Smits R. Blocking Wnt Secretion Reduces Growth of Hepatocellular Carcinoma Cell Lines Mostly Independent of β-Catenin Signaling. Neoplasia. 2016 Dec;18(12):711-723.

3) Angelika Bröer, Stephen Fairweather, Stefan Bröer. Disruption of Amino Acid Homeostasis by Novel ASCT2 Inhibitors Involves Multiple Targets. Front Pharmacol. 2018; 9: 785

4) Cruzat V, Macedo Rogero M, Noel Keane K, Curi R, Newsholme P. Glutamine: Metabolism and Immune Function, Supplementation and Clinical Translation. Nutrients. 2018 Oct 23;10(11):1564.

5) Le A, Lane AN, Hamaker M, Bose S, Gouw A, Barbi J, Tsukamoto T, Rojas CJ, Slusher BS, Zhang H, Zimmerman LJ, Liebler DC, Slebos RJ, Lorkiewicz PK, Higashi RM, Fan TW, Dang CV. Glucose-independent glutamine metabolism via TCA cycling for proliferation and survival in B cells. Cell Metab. 2012 Jan 4;15(1):110-21.